# Malignancies in Patients with Interstitial Lung Diseases: A Single Center Observational Study

**DOI:** 10.3390/jcm11247321

**Published:** 2022-12-09

**Authors:** Haishuang Sun, Min Liu, Xiaoyan Yang, Yanhong Ren, Bingbing Xie, Jing Geng, Mei Deng, Huaping Dai, Chen Wang

**Affiliations:** 1Department of Respiratory Medicine, The First Hospital of Jilin University, Changchun 130021, China; 2National Center for Respiratory Medicine, National Clinical Research Center for Respiratory Diseases, Institute of Respiratory Medicine, Chinese Academy of Medical Sciences, Department of Pulmonary and Critical Care Medicine, Center of Respiratory Medicine, China-Japan Friendship Hospital, Beijing 100029, China; 3Chinese Academy of Medical Sciences and Peking Union Medical College, Beijing 100730, China; 4Department of Radiology, China-Japan Friendship Hospital, Beijing 100029, China

**Keywords:** interstitial lung disease, malignancy, lung cancer, prevalence

## Abstract

Objective: Current studies focus on the prevalence rate of lung cancer in idiopathic pulmonary fibrosis and connective tissue disease-associated interstitial lung disease (CTD-LID). Our aim was to investigate the prevalence of malignancies in patients with various subtypes of ILD. Methods: A total of 5350 patients diagnosed with ILD between January 2015 and December 2021 were retrospectively included. The prevalence of different malignancies and different ILDs was assessed using complete follow-up data. Results: A total of 248 patients (139 males; 65—IQR, 57 to 72—years) out of 5350 patients with ILD were confirmed with malignancies. A total of 69% of patients with ILD and malignances were older than 60 years old. The prevalence of malignancies in ILD patients was 4.6%, and lung cancer had the most common incidence of 1.9%, followed by malignancies in the digestive system of 0.9%. Among the different ILD subtypes, the prevalence of malignancies such as organizing pneumonia (OP), idiopathic pulmonary fibrosis (IPF), anti-neutrophil cytoplasmic antibodies-associated vasculitis-related ILD(AAV-ILD), nonspecific interstitial pneumonia (NSIP), CTD-ILD, hypersensitivity pneumonitis (HP), sarcoidosis, and other types of ILD was 6.8%, 5.0%, 4.7%, 4.3%, 2.5%, 2.2%, 1.2%, and 6.9%, respectively. The incidence of lung cancer as the most common tumor in IPF was 3.9%, with adenocarcinoma predominating (1.7%). The highest rate of malignancy occurring in RA of CTD-ILD was 2.4%. Conclusion: Older patients with ILD (≥60 years) including OP, IPF, AAV-ILD, NSIP, CTD-ILD, and HP, were associated with a higher incidence of malignancy, especially males aged from 60 to 69 years. These epidemiological results indicate that it is essential for physicians to pay more attention to the screening of and management strategies for different malignancies, according to the specific ILD subtypes.

## 1. Introduction

Interstitial lung diseases (ILDs) are a group of heterogeneous diseases caused by various types of alveolar inflammation and/or fibrosis, including idiopathic interstitial pneumonias (IIPs), collagen vascular disease-associated interstitial pneumonia, and hypersensitivity pneumonia (HP). Idiopathic pulmonary fibrosis (IPF) represents the most common IIP and is one of the most aggressive ILDs. The available evidence suggested an increased risk of lung cancer, systemic sclerosis, and certain forms of pneumoconiosis in IPF [1,2]. The prevalence of lung cancer in patients with IPF ranges from 3.3% to 48%, and several studies have demonstrated that the prevalence of lung cancer is 7.3–3.34 times higher in patients with IPF compared to the general population [3,4,5]. Moreover, several studies have reported that connective tissue disease-associated interstitial lung disease (CTD-ILD) is associated with a higher incidence of malignancies [6,7,8]. An increased incidence of lymphoma is detected in Sjögren’s syndrome (SS). Moreover, patients with ILD combined with malignancy have a worse prognosis, compared to those with ILD alone [1,4,8]. Investigating the prevalence of multiple malignancies in various types of ILD is beneficial for the early detection of malignancies in the follow-up of ILD. However, previous studies have focused on the occurrence of lung cancer in ILD, especially IPF and CTD-ILD, reports on other malignancies are rare. Therefore, the aim of our study was to fully investigate the incidence of different malignancies in various ILD subtypes and to provide some information for future clinical surveillance.

## 2. Materials and Methods

### 2.1. Study Cohort and Design

The study was approved by the ethics committee of our Hospital (No.: 2022-KY-031). Informed consent was waived because the study was retrospective. Data from the national clinical research center for respiratory diseases for ILD were collected from January 2015 to December 2021. First, patients diagnosed with ILD were screened from an electronic medical record system. ILDs were classified into the following classes, according to the American Thoracic Society/European Respiratory Society guidelines [5] including IPF, non-specific interstitial pneumonia (NSIP), COP, respiratory bronchiolitis-associated ILD (RB-ILD), lymphocytic interstitial pneumonia (LIP), desquamative interstitial pneumonitis (DIP), acute interstitial pneumonia (AIP), pleuroparenchymal fibroelastosis, CTD-ILD, HP, sarcoidosis, and anti-neutrophil cytoplasmic antibodies-associated vasculitis (AAV) -related ILD (AAV-ILD). Among these, CTD-ILD was further categorized into rheumatoid arthritis related ILD (RA-ILD), idiopathic inflammatory myopathies related ILD (IIM-ILD), SS related ILD (SS-ILD), systemic lupus erythematosus related ILD (SLE-ILD), systemic sclerosis related ILD (SSc-ILD), and overlap syndrome (OS)- related ILD (OS-ILD).

Information on patients, including gender, age, time of diagnosis, and comorbidities including hypertension, diabetes, coronary heart disease, venous thromboembolism (VTE) and other comorbidities were collected in this study. The age-adjusted Charlson comorbidity index (ACCI) for oncology patients, which is a combination of the age equivalence index and CCI [9], was also analyzed in our study. The primary endpoint was the incidence of malignancy. The exclusion criteria included: (1) patients under 18 years of age; (2) patients without a chest CT in our center; (3) incomplete follow-up information; and (4) a time of follow-up of less than 6 months. Figure 1 illustrates the flow chart.

### 2.2. Statistical Analysis

All statistical analyses were performed using SPSS software (version 24.0, IBM Corporation, Armonk, NY, USA). We collected information on patients diagnosed with ILD from 2015 to 2021 and calculated the prevalence (%) of different malignancies. Data were expressed as the median and interquartile range (IQR) or mean ± standard deviation (SD). Mann–Whitney U tests were used for the analysis of differences between groups. Fisher’s exact test or chi-square tests were used to assess categorical variables.

## 3. Results

### 3.1. Demographic Characteristics of Patients with ILD

A total of 5350 eligible patients with ILD, including IIP, CTD-ILD, and other types of ILD, were included in this research (Table 1). The median age of the patients with ILD was 61 years (IQR, 52 to 70 years), and people over 60 years old tended to have a higher incidence of the disease. The proportion of males and females was similar (50.4% vs. 49.6%). Patients with ILD frequently presented with various complications (hypertension, 24.6%; diabetes, 15.9%; coronary heart disease, 12.0%; venous thromboembolism, 2.5%; other complications, 51.2%, respectively) (Table 1). In patients with IIP, those with IPF were generally older than those presenting with the other subtypes, with a median age of 67 years (IQR, 61 to 72 years), and exhibited a higher incidence of hypertension, diabetes, coronary artery disease, VTE, and other comorbidities at a rate of 23.9%, 23.3%, 22.2%, 3.5%, and 38.0%, respectively. IPF and OP were the most common, accounting for 33.0% and 34.5% of IIP, respectively (Table 2). In the CTD-ILD population, IIM was the most common, accounting for 12.3% of ILDs (Figure 2). Those belonging to the different subtypes were all predominantly male, and SS and RA showed a higher incidence of hypertension at 23.7% and 22.6%, respectively. Compared with other CTD-ILDs, RA showed a higher incidence of diabetes, coronary heart disease, and VTE at 17.1%, 10.7%, and 3.6%, respectively (Table 3).

The analysis of the proportion of each ILD subtype in the total ILD population from 2015 to 2021 demonstrated an increasing trend in the proportion of CTD-ILD, but a slow decrease in the proportion of OP. The proportion of IPF was relatively stable (Figure 3).

### 3.2. Prevalence of Malignancies in Patients with ILD

The prevalence of malignancies among patients with ILD was 4.6%, among which lung cancer was the most common at 1.9% (of which adenocarcinoma accounted for the highest percentage at 1.0%), followed by digestive system tumors (0.9%), breast cancer (0.6%), lymphoma (0.4%), genitourinary system tumor (0.4%), thyroid cancer (0.3%), head and neck tumors (0.1%), and other tumors (0.1%). Figure 4 shows cases with ILD and malignancies. Compared to CTD-ILD, IIP showed a higher incidence of malignancy of 5.3% and 2.5%, respectively. Lung cancer had the highest significant incidence of 2.6% among IIP patients. Whereas, among CTD-ILD, lung cancer, breast cancer, thyroid cancer, and digestive system tumor had similar incidence rates of 0.6%, 0.8%, 0.5%, and 0.5%, respectively. In addition, adenocarcinoma made up the highest percentage of IIP compared to CTD-ILD and other types of ILD with 1.5%, 0.3%, and 1.1%, respectively (Table 4). In order of malignancy incidence among IIP patients, lung cancer had the highest incidence at 3.9% and 2.0% in IPF and NSIP, respectively, while breast cancer was the most common in OP, with an incidence of 2.2%, followed by lung cancer, with an incidence of 1.8%. Meanwhile, among the different types of IIP, lung adenocarcinoma was the most common pathological type in IPF, NSIP, and OP, accounting for 1.7%, 1.5%, and 1.3%, respectively (Table 5 and Figure 5). Regarding the CTD-ILD subtypes, RA-ILD, IIM-ILD, and SS-ILD had similar malignancy incidence rates of 2.4%, 2.3%, and 2.3%, respectively, while OS-ILD, SLE-ILD, and SSc-ILD were all 0%. Among other ILD subtypes, AAV-ILD, HP, and sarcoidosis exhibited higher malignancy rates of 4.7%, 2.2%, and 1.2%, respectively (Table 6 and Figure 5).

The incidence of malignancies in ILD gradually increased from 2.1% in 2015 to 5.9% in 2021. Similarly, the prevalence of lung cancer in ILD increased from 0.4% in 2015 to 2.6% in 2021. The prevalence of other malignancies in ILD did not change significantly over time (Figure 6).

### 3.3. Demographics Characteristics of ILD Developed Malignancies

The median age of patients with ILD combined with malignancies was 65 years (IQR, 57 to 72 years). Patients with digestive system tumors had a relatively older age of 68 years (IQR, 61–75 years), and patients with thyroid cancer were younger at 46 years (IQR, 39–62 years). The majority of patients were over 60 years of age (69.0%) and predominantly male (all tumors, 56.0%; lung cancer, 75.2%; lymphoma, 59.1%; digestive system malignant tumor, 60.9%; head and neck malignant tumor, 100.0%, respectively). Apart from other complications (44.0%), hypertension (24.6%) was the most common complication among patients, followed by diabetes (13.3%), coronary heart disease (10.1%), and VTE (2.8%) (Table 7).

In the population with ILD combined with lung cancer, adenocarcinoma was the most common in IIP, accounting for 58.5%, and predominantly in the right lung, at 53.7%. With the exception of VTE (IIP, 4.9%; non-IIP, 3.3%), various other complications occurred more frequently in non-IIP patients, although no statistical differences were observed (Table 8).

## 4. Discussion

To our knowledge, this is the first study providing the prevalence of different malignancies in various subtypes of ILDs. The prevalence of malignancies in ILD was 4.6%, of which lung cancer made up 1.9%. The prevalence of malignancies in different ILD subtypes, including OP, IPF, AAV-ILD, NSIP, CTD-ILD, HP, and sarcoidosis, respectively, was 6.8%, 5.0%, 4.7%, 4.3%, 2.5%, 2.2%, and 1.2%.

IPF has been proved to have an extensive epidemiological and pathogenetic association with malignancy [1,10,11,12]. Pulmonary fibroblasts share many similar characteristics with cancer cells, including increased cell proliferation rate, senescence, resistance to apoptosis, and telomere wear. Additionally, genetic and epigenetic alterations, altered intercellular communication, abnormal activation of signal transduction pathways, and tissue invasion are all etiologic similarities between pulmonary fibrosis and malignancy [13,14,15]. Several studies have revealed that IPF is an essential risk factor for lung cancer [16,17], and the prevalence of lung cancer in IPF ranges from 3.3% to 48% [4,18,19,20]. The prevalence of lung cancer in IPF patients in this study was 3.9%, predominantly adenocarcinoma, which was similar to the results found in the previous data. The majority of studies focused only on the occurrence of lung cancer; the prevalence of IPF combined with all malignancies, including tumors of the digestive and urinary systems et al., was 5.0% in the present study. Our data demonstrated a prevalence of 2.5% for malignancies in CTD-ILD, of which lung cancer remained the most common type, and the patients were more inclined to be female patients younger than 60 years old, which is similar to the results of previous studies [8,20]. As one of the major subtypes of CTD-ILD, the relationship between IIM-ILD, including dermatomyositis (DM) and polymyositis (PM), and malignancy has been extensively studied. This association is more pronounced in DM than in PM. The prevalence of DM-associated malignancy was highly variable, ranging from less than 7% to more than 30% [21]. Ovarian or breast cancer in women and lung cancer in men predominate [22]. Malignancies occurred in 2.3% of IIM-ILD in our research, with lung cancer being predominant. The lower malignancy incidence may be related to the differences in the included population and insufficient follow-up time. Epidemiological studies of OP combined with malignancy were less reported, and a high incidence of 6.8% was shown in this research, of which breast cancer was predominant (2.2%), followed by lung cancer (1.8%), with lung lesion after malignancy treatment considered as the main cause. In addition, a growing amount of evidence supports the association between AAV-ILD and malignancy, and the prevalence of malignancy ranged from 2.7% to 26% in AAV [23,24,25]. The most commonly reported malignancies were non-melanoma skin carcinomas, bladder cancer, lung cancer, prostate cancer, breast cancer, and colorectal cancer [23,24,25,26]. Similarly, the incidence of malignancy in AAV- ILD in our study was 4.7%. We also introduced the novel ACCI scoring system, which was first proposed by Charlson et al., for predicting the incidence of perioperative complications [9]. Although the mean ACCI score was slightly higher in the non-IIP group than in the IIP group, there was no statistically significant difference between the two groups.

The use of glucocorticoids, biologics, and immunosuppressive agents such as methotrexate in ILD can inhibit tumor necrosis factor, which may be related to the development of malignancy [27]. Previous studies of malignancies in OP are scarce, given the extensive research into the mechanisms of glucocorticoid therapy and the development of malignancies, the explanation that glucocorticoid therapy in OP may cause immune changes in the body, and thus increase the incidence of malignancies is not excluded. Close follow-up and early identification of and intervention for malignancy in such patients is particularly essential to improve patient survival. The widespread prevalence of different malignancies in various ILD subtypes in multiple studies may be related to different study populations, diagnostic criteria, and study designs.

There are still several limitations to this research. First, although this is a large demographic study, as a retrospective single-center study, the patients may not adequately represent the ILD and malignancy populations. Second, patients who were lost to follow-up were excluded from our study, which may have contributed to selection bias. Finally, the confounding effects of smoking history on the development of malignancy were not eliminated from the research.

## 5. Conclusions

Patients with ILD, including OP, IPF, NSIP, CTD-ILD, and HP, have a higher prevalence rate of malignancy, especially older male patients. In patients with CTD-ILD, compared to other causes, AAV- ILD has a higher prevalence of malignancies, especially lung cancer, thyroid cancer, and breast cancer. Based on the above results, the clinicians should develop precise and personalized follow-up strategies according to the different malignancy rates of patients with different types of ILD to improve diagnostic efficiency.

## Figures and Tables

**Figure 1 jcm-11-07321-f001:**
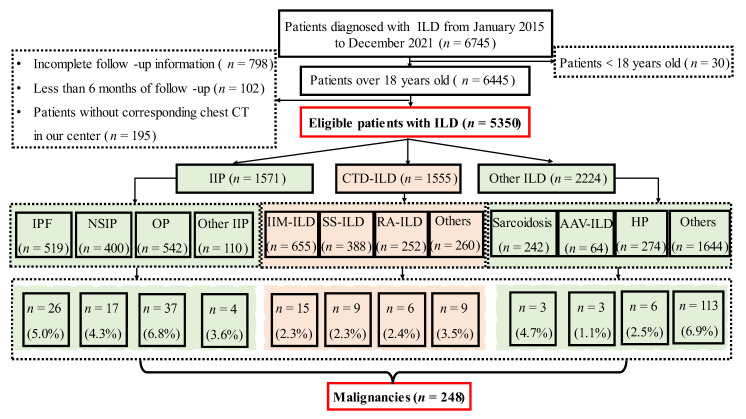
Study Design Flow Chart. ILD, interstitial lung disease; IPF, idiopathic pulmonary fibrosis; NSIP, nonspecific interstitial pneumonia; OP, organizing pneumonia; CTD-ILD, connective tissue disease-associated ILD; HP, hypersensitivity pneumonitis; AAV, anti-neutrophil cytoplasmic antibodies-associated vasculitis; IIM, idiopathic inflammatory myopathies; SS, Sjögren’s syndrome; RA, rheumatoid arthritis.

**Figure 2 jcm-11-07321-f002:**
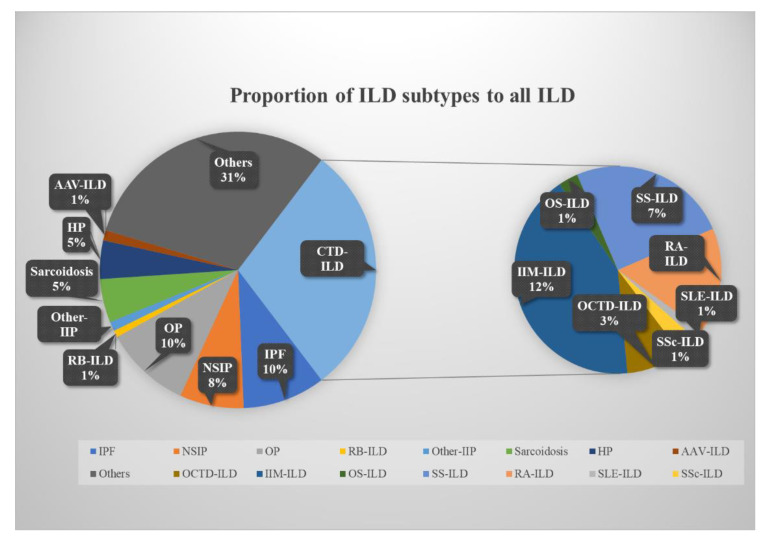
Percentage of ILD subtypes compared to the total ILD population. ILD, interstitial lung disease; IPF, idiopathic pulmonary fibrosis; NSIP, nonspecific interstitial pneumonia; OP, organizing pneumonia; CTD-ILD, connective tissue disease-associated ILD; HP, hypersensitivity pneumonitis; AAV, anti-neutrophil cytoplasmic antibodies-associated vasculitis; IIM, idiopathic inflammatory myopathies; OS, overlap syndrome; SS, Sjögren’s syndrome; RA, rheumatoid arthritis; SLE, systemic lupus erythematosus; SSc, systemic sclerosis; OCTD-ILD, other CTD-ILD.

**Figure 3 jcm-11-07321-f003:**
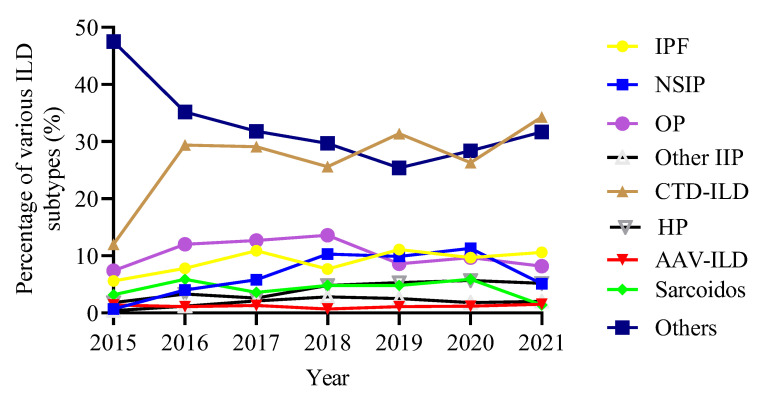
Trends in the proportion of ILD subtypes over time. ILD, interstitial lung disease; IPF, idiopathic pulmonary fibrosis; NSIP, nonspecific interstitial pneumonia; OP, organizing pneumonia; CTD-ILD, connective tissue disease-associated ILD; HP, hypersensitivity pneumonitis; AAV, anti-neutrophil cytoplasmic antibodies-associated vasculitis.

**Figure 4 jcm-11-07321-f004:**
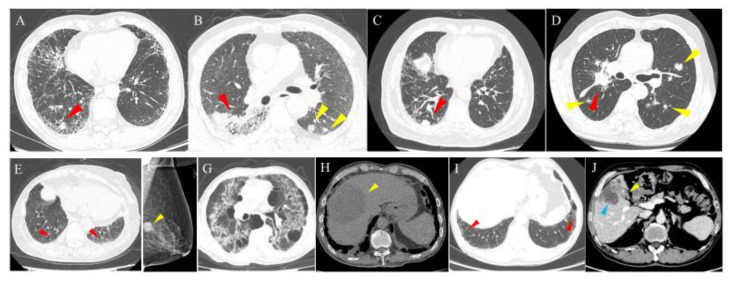
Cases of patients with ILD and malignancies. (**A**) A 65-year-old male with IPF and lung cancer. HRCT shows an irregular solid nodule in the right lower lobe (red arrowhead). (**B**) A 62-year-old male with NSIP and lung cancer. HRCT shows an irregular mass and ground glass opacity (red arrowhead) in the right lower lobe and multiple irregular nodules in the left lower lobe (yellow arrowhead). (**C**) A 74-year-old male with RA-associated ILD and lung cancer. HRCT shows a subpleural solid nodule in the right lower lung (red arrowhead). (**D**) A 63-year-old male with OP and lung cancer. HRCT shows an irregular mass in the right hilum (red arrowhead) and multiple metastases in bilateral lungs (yellow arrowhead). A 90-year-old female patient with ILD and breast cancer. HRCT shows ground glass opacity and a reticular pattern (red arrowhead) in the bilateral lower lobes (**E**) and a dense nodule (yellow arrowhead) in the left breast on mammography (**F**). A 57-year-old male patient with ILD and hepatocellular carcinoma. HRCT shows multiple patches of ground glass opacity, a reticular pattern, cystic spaces, and traction bronchiectasis in the bilateral lungs (**G**), and abdominal non-contrasted CT shows hepatocellular carcinoma (yellow arrowhead) as an irregular lower density mass in the right lobe of the liver (**H**). A 65-year-old male patient with ILD combined with gallbladder cancer. HRCT shows subpleural ground glass opacity and a reticular pattern (red arrowhead) in the bilateral lower lungs (**I**). Abdominal contrasted CT shows the enhanced lesion in the gallbladder (yellow arrowhead) and liver invasion (blue arrowhead) (**J**). HRCT, high-resolution computed tomography; ILD, interstitial lung disease; IPF, idiopathic pulmonary fibrosis; NSIP, nonspecific interstitial pneumonia; RA, rheumatoid arthritis; OP, organizing pneumonia.

**Figure 5 jcm-11-07321-f005:**
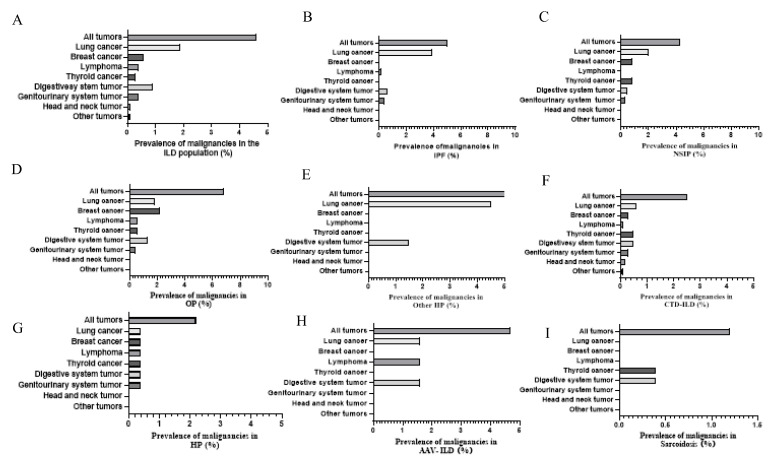
Incidence of malignancies in different ILD subgroups. (**A**) All subtypes of ILD. (**B**) Idiopathic pulmonary fibrosis. (**C**) Nonspecific interstitial pneumonia. (**D**) Organizing pneumonia. (**E**) Other IIP. (**F**) Connective tissue disease-associated ILD. (**G**) Hypersensitivity pneumonitis. (**H**) Anti-neutrophil cytoplasmic antibodies-associated vasculitis related-ILD. (**I**) Sarcoidosis. ILD, interstitial lung disease.

**Figure 6 jcm-11-07321-f006:**
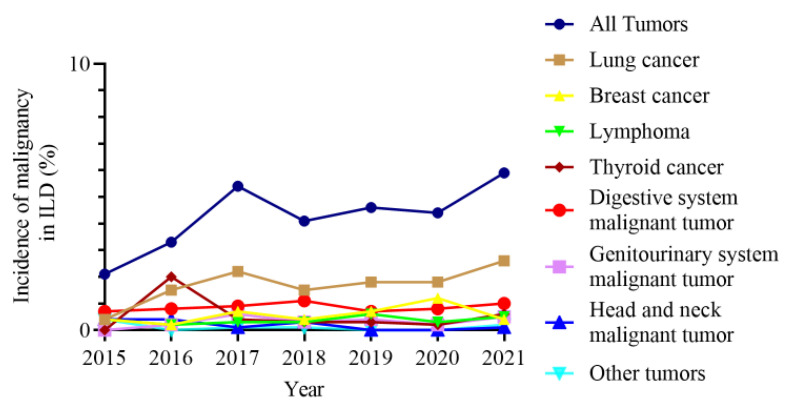
Prevalence of malignancies in ILD over time. ILD, interstitial lung disease.

**Table 1 jcm-11-07321-t001:** Total demographic characteristics of patients with IIP, CTD-ILD, and other ILDs.

Characteristics	Total *n* = 5350 (%)	IIP *n* = 1571 (%)	CTD-ILD *n* = 1555 (%)	Others *n* = 2224 (%)
Age, Median (IQR) years	61 (52–70)	63 (55–70)	59 (50–67)	62 (52–71)
<60	2146 (43.9)	606 (38.6)	791 (50.9)	952 (42.8)
60–69	1654 (30.9)	538 (34.2)	480 (30.9)	636 (38.5)
70–79	1010 (18.9)	345 (22.0)	237 (15.2)	428 (19.2)
≥80	337 (6.3)	82 (5.4)	47 (3.0)	208 (9.4)
Male	2697 (50.4)	986 (62.8)	485 (31.2)	1226 (55.1)
Comorbidities				
Hypertension	1317 (24.6)	397 (25.3)	308 (19.8)	612 (27.5)
Diabetes	849 (15.9)	290 (18.5)	208 (13.4)	351 (15.8)
Coronary heart disease	641 (12.0)	228 (14.5)	113 (7.3)	300 (13.5)
Venous thromboembolism	127 (2.5)	41 (2.6)	47 (3.0)	39 (1.8)
Other comorbidities	2738 (51.2)	532 (33.9)	1460 (93.9)	746 (33.5)

IIP, idiopathic interstitial pneumonia; CTD-ILD, connective tissue disease-associated ILD; IQR, interquartile range.

**Table 2 jcm-11-07321-t002:** Demographic characteristics of IIP.

Characteristics	Total *n* = 1571 (%)	IPF *n* = 519 (%)	NSIP *n* = 400 (%)	OP *n* = 542 (%)	RB-ILD *n* = 44 (%)	Other IIPs *n* = 66 (%)
Age, Median (IQR) years	63 (55–70)	67 (61–72)	61 (53–69)	60 (52–68)	56 (49–64)	59 (48–68)
<60	606 (38.6)	101 (19.5)	182 (45.5)	261 (48.2)	28 (63.6)	34 (51.5)
60–69	538 (34.2)	215 (41.4)	128 (32.0)	166 (30.6)	10 (22.7)	19 (28.8)
70–79	345 (22.0)	163 (31.4)	76 (19.0)	91 (16.8)	59 (11.4)	10 (15.2)
≥80	82 (5.2)	40 (7.7)	14 (3.5)	24 (4.4)	1 (2.3)	3 (4.5)
Male	986 (62.8)	432 (83.2)	197 (49.3)	274 (50.6)	40 (90.9)	43 (65.2)
Comorbidities						
Hypertension	397 (25.3)	124 (23.9)	101 (25.3)	146 (26.9)	11 (25.0)	15 (22.7)
Diabetes	290 (18.5)	121 (23.3)	57 (14.2)	98 (18.1)	7 (15.9)	7 (10.6)
Coronary heart disease	228 (14.5)	115 (22.2)	58 (14.5)	44 (8.1)	4 (9.1)	7 (10.6)
Venous thromboembolism	41 (2.6)	18 (3.5)	10 (2.5)	13 (2.4)	0 (0)	0 (0)
Other comorbidities	532 (33.9)	197 (38.0)	114 (28.5)	191 (35.2)	3 (6.8)	27 (40.9)

IIP, idiopathic interstitial pneumonia; ILD, interstitial lung disease; IPF, idiopathic pulmonary fibrosis; NSIP, nonspecific interstitial pneumonia; OP, organizing pneumonia; RB-ILD, respiratory bronchiolitis-associated ILD; IQR, interquartile range.

**Table 3 jcm-11-07321-t003:** Demographic characteristics of ILD (except IIP).

Characteristics	CTD–ILD *n* = 1555 (%)	IIM-ILD *n* = 655 (%)	OS-ILD *n* = 50 (%)	SS-ILD *n* = 388 (%)	RA-ILD *n* = 252 (%)	SLE-ILD *n* = 23 (%)	SSc-ILD *n* = 50 (%)	OCTD–ILD *n* = 137 (%)	HP *n* = 274 (%)	Sarcoidosis *n* = 242 (%)	AAV-ILD, *n* = 64 (%)	Others *n* = 1644 (%)
Age, Median (IQR) years	59 (50–67)	54 (46–62)	53 (44–60)	64 (57–70)	65 (59–73)	41 (32–50)	55 (49–63)	62 (52–70)	58 (47–66)	54 (44–60)	68 (59–75)	64 (54–73)
<60	791 (50.9)	449 (68.5)	37 (74.0)	123 (31.7)	71 (28.2)	21 (91.3)	31 (62.0)	59 (43.1)	126 (52.1)	203 (74.1)	16 (25.0)	607 (36.9)
60–69	480 (30.9)	162 (24.7)	12 (24.0)	164 (42.3)	87 (34.5)	1 (4.3)	14 (28.0)	40 (29.2)	79 (32.6)	57 (20.8)	21 (32.8)	479 (29.1)
70–79	237 (15.2)	39 (6.0)	1 (2.0)	88 (22.7)	74 (29.4)	1 (4.3)	5 (10.0)	29 (21.2)	35 (14.5)	11 (4.0)	19 (29.7)	363 (22.1)
≥80	47 (3.0)	5 (0.8)	0 (0)	13 (3.4)	20 (7.9)	0 (0)	0 (0)	9 (6.6)	2 (0.8)	3 (1.1.)	8 (12.5)	195 (11.9)
Male	485 (31.2)	181 (27.6)	7 (14.0)	132 (34.0)	100 (39.7)	0 (0)	3 (6.0)	62 (45.3)	112 (46.3)	74 (27.0)	35 (54.7)	2697 (50.4)
Comorbidities												
Hypertension	308 (19.8)	119 (18.2)	6 (12.0)	92 (23.7)	57 (22.6)	3 (13.0)	1 (2.0)	30 (21.9)	65 (26.9)	64 (23.4)	13 (20.3)	470 (28.6)
Diabetes	208 (13.4)	95 (14.5)	2 (4.0)	51 (13.1)	43 (17.1)	1 (4.3)	1 (2.0)	15 (10.9)	30 (12.4)	37 (13.5)	10 (15.6)	274 (16.7)
Coronary heart disease	113 (7.3)	34 (5.2)	2 (4.0)	39 (10.1)	27 (10.7)	0 (0)	2 (4.0)	9 (6.6)	24 (9.9)	6 (2.2.)	6 (9.4)	264 (16.1)
Venous thromboembolism	47 (3.0)	27 (4.1)	0 (0)	5 (1.3)	9 (3.6)	2 (8.7)	0 (0)	4 (2.9)	8 (3.3)	0 (0)	5 (7.8)	26 (1.6)
Other comorbidities	1460 (93.9)	655 (100.0)	50 (100.0)	388 (100.0)	252 (100.0)	23 (100.0)	50 (100.0)	42 (30.7)	104 (43.0)	0 (0)	64 (100.0)	628 (38.2)

CTD-ILD, connective tissue disease-associated ILD; IIM, idiopathic inflammatory myopathies; OS, overlap syndrome; SS, Sjogren syndrome; RA, rheumatoid arthritis; SLE, systemic lupus erythematosus; SSc, systemic sclerosis; HP, hypersensitivity pneumonitis; AAV, anti-neutrophil cytoplasmic antibodies-associated vasculitis; OCTD-ILD, other CTD-ILD.

**Table 4 jcm-11-07321-t004:** Malignancies in IIP, CTD-ILD, and other ILDs.

Characteristics	Total *n* = 5350 (%)	IIP *n* = 1571 (%)	CTD-ILD *n* = 1555 (%)	Others *n* = 2224 (%)
All tumors	248 (4.6)	84 (5.3)	39 (2.5)	125 (5.6)
Lung cancer	101 (1.9)	41 (2.6)	10 (0.6)	50 (2.2)
Adenocarcinoma	54 (1.0)	24 (1.5)	5 (0.3)	25 (1.1)
Squamous cell carcinoma	29 (0.5)	7 (0.5)	3 (0.2)	19 (0.8)
Others	18 (0.3)	10 (0.6)	2 (0.1)	6 (0.3)
Breast cancer	31 (0.6)	15 (0.3)	12 (0.8)	12 (0.6)
Lymphoma	22 (0.4)	4 (0.3)	2 (0.1)	16 (0.7)
Thyroid cancer	18 (0.3)	6 (0.4)	7 (0.5)	5 (0.2)
Digestive system tumor	46 (0.9)	13 (0.8)	8 (0.5)	25 (1.1)
Genitourinary system tumor	19 (0.4)	5 (0.3)	4 (0.3)	10 (0.4)
Head and neck tumor	6 (0.1)	0 (0)	3 (0.2)	3 (0.1)
Other tumors	5 (0.1)	0 (0)	1 (0.1)	4 (0.2)

ILD, interstitial lung disease; IIP, idiopathic interstitial pneumonia; CTD-ILD, connective tissue disease-associated ILD.

**Table 5 jcm-11-07321-t005:** Malignancies in various subtypes of IIP.

Characteristics	Total *n* = 1571 (%)	IPF *n* = 519 (%)	NSIP *n* = 400 (%)	OP *n* = 542 (%)	RB-ILD *n* = 44 (%)	Other IIP *n* = 66 (%)
All tumors	84 (5.3)	26 (5.0)	17 (4.3)	37 (6.8)	0 (0)	4 (6.1)
Lung cancer	41 (2.6)	20 (3.9)	8 (2.0)	10 (1.8)	0 (0)	3 (4.5)
Adenocarcinoma	24 (1.5)	9 (1.7)	6 (1.5)	7 (1.3)	0 (0)	2 (3.0)
Squamous cell carcinoma	7 (0.4)	5 (1.0)	0 (0)	1 (0.2)	0 (0)	1 (1.5)
Others	10 (0.6)	6 (1.2)	2 (0.5)	2 (0.4)	0 (0)	0 (0)
Breast cancer	15 (1.0)	0 (0)	3 (0.8)	12 (2.2)	0 (0)	0 (0)
Lymphoma	4 (0.3)	1 (0.2)	0 (0)	3 (0.6)	0 (0)	0 (0)
Thyroid cancer	6 (0.4)	0 (0)	3 (0.8)	3 (0.6)	0 (0)	0 (0)
Digestive system tumor	13 (0.8)	3 (0.6)	2 (0.5)	7 (1.3)	0 (0)	1 (1.5)
Genitourinary system tumor	5 (0.3)	2 (0.4)	1 (0.3)	2 (0.4)	0 (0)	0 (0)
Head and neck tumor	0 (0)	0 (0)	0 (0)	0 (0)	0 (0)	0 (0)
Other tumors	0 (0)	0 (0)	0 (0)	0 (0)	0 (0)	0 (0)

IIP, idiopathic interstitial pneumonia; ILD, interstitial lung disease; IPF, idiopathic pulmonary fibrosis; NSIP, nonspecific interstitial pneumonia; OP, organizing pneumonia; RB-ILD, respiratory bronchiolitis-associated ILD.

**Table 6 jcm-11-07321-t006:** Demographic characteristics and prevalence of malignancies in CTD-ILD and other types of ILDs.

Characteristics	IIM-ILD *n* = 655 (%)	OS-ILD *n* = 50 (%)	SS-ILD *n* = 388 (%)	RA-ILD *n* = 252 (%)	SLE-ILD *n* = 23 (%)	SSc-ILD *n* = 50 (%)	OCTD-ILD *n* = 137 (%)	Sarcoidosis *n* = 242 (%)	AAV-ILD *n* = 64 (%)	HP *n* = 274 (%)	Others *n* = 1644 (%)
All tumors	15 (2.3)	0 (0)	9 (2.3)	6 (2.4)	0 (0)	0 (0)	9 (6.6)	3 (1.2)	3 (4.7)	6 (2.2)	113 (6.9)
Lung cancer	4 (0.6)	0 (0)	0 (0)	4 (1.6)	0 (0)	0 (0)	2 (1.5)	0 (0)	1 (1.6)	1 (0.4)	48 (2.9)
Adenocarcinoma	1 (0.2)	0 (0)	0 (0)	2 (0.8)	0 (0)	0 (0)	2 (1.5)	0 (0)	0 (0)	1 (0.4)	24 (1.5)
Squamous cell carcinoma	2 (0.3)	0 (0)	0 (0)	1 (0.4)	0 (0)	0 (0)	0 (0)	0 (0)	0 (0)	0 (0)	19 (1.2)
Others	1 (0.2)	0 (0)	0 (0)	1 (0.4)	0 (0)	0 (0)	0 (0)	0 (0)	1 (1.6)	0 (0)	5 (0.3)
Breast cancer	2 (0.3)	0 (0)	0 (0)	0 (0)	0 (0)	0 (0)	2 (1.5)	0 (0)	0 (0)	1 (0.4)	11 (0.7)
Lymphoma	1 (0.2)	0 (0)	1 (0.3)	0 (0)	0 (0)	0 (0)	0 (0)	0 (0)	1 (1.6)	1 (0.4)	14 (0.9)
Thyroid cancer	4 (0.6)	0 (0)	1 (0.3)	1 (0.4)	0 (0)	0 (0)	1 (0.7)	1 (0.4)	0 (0)	1 (0.4)	3 (0.2)
Digestive system tumor	2 (0.3)	0 (0)	5 (1.3)	1 (0.4)	0 (0)	0 (0)	0 (0)	1 (0.4)	1 (1.6)	1 (0.4)	22 (1.3)
Genitourinary system tumor	2 (0.3)	0 (0)	2 (0.5)	0 (0)	0 (0)	0 (0)	0 (0)	0 (0)	0 (0)	1 (0.4)	9 (0.5)
Head and neck tumor	0 (0)	0 (0)	0 (0)	0 (0)	0 (0)	0 (0)	3 (2.2)	0 (0)	0 (0)	0 (0)	3 (0.2)
Other tumors	0 (0)	0 (0)	0 (0)	0 (0)	0 (0)	0 (0)	1 (0.7)	1 (0.4)	0 (0)	0 (0)	3 (0.2)

ILD, interstitial lung disease; CTD-ILD, connective tissue disease-associated ILD; IIM, idiopathic inflammatory myopathies; OS, overlap syndrome; SS, Sjögren’s syndrome; RA, rheumatoid arthritis; SLE, systemic lupus erythematosus; SSc, systemic sclerosis; OCTD-ILD, other CTD-ILD; AAV, anti-neutrophil cytoplasmic antibodies-associated vasculitis; HP, hypersensitivity pneumonitis.

**Table 7 jcm-11-07321-t007:** Demographic characteristics of patients with ILD-developed malignancies.

Characteristics	All Tumors *n* = 248 (%)	Lung Cancer *n* = 101 (%)	Breast Cancer *n* = 31 (%)	Lymphoma *n* = 22 (%)	Thyroid Cancer *n* = 18 (%)	Digestive System Tumor *n* = 46 (%)	Genitourinary System Tumor *n* = 19 (%)	Head and Neck Tumor *n* = 6 (%)	Other Tumors *n* = 5 (%)
Age, Median (IQR) years	65 (57–72)	66 (60–71)	56 (52–72)	65 (57–73)	46 (39–62)	68 (61–75)	66 (60–76)	64 (62–72)	62 (59–72)
<60	77 (31.0)	25 (24.8)	16 (51.6)	7 (31.8)	12 (66.7)	11 (23.9)	4 (21.1)	0 (0)	2 (40.0)
60–69	94 (37.9)	43 (42.6)	6 (19.4)	8 (36.4)	4 (22.2)	18 (39.1)	8 (42.1)	5 (83.3)	2 (40.0)
70–79	60 (24.2)	29 (28.7)	6 (19.4)	7 (31.8)	2 (11.1)	11 (23.9)	5 (26.3)	0 (0)	0 (0)
≥80	17 (6.9)	4 (4.0)	3 (9.7)	0 (0)	0 (0)	6 (13.0)	2 (10.5)	1 (16.7)	1 (20.0)
Male	139 (56.0)	76 (75.2)	0 (0)	13 (59.1)	4 (22.2)	28 (60.9)	9 (47.4)	6 (100.0)	3 (60.0)
Comorbidities									
Hypertension	61 (24.6)	24 (23.8)	4 (12.9)	9 (40.9)	4 (22.2)	8 (17.4)	8 (42.1)	3 (50.0)	1 (20.0)
Diabetes	33 (13.3)	15 (14.9)	1 (3.2)	3 (13.6)	0 (0)	7 (15.2)	6 (31.6)	1 (16.7)	0 (0)
Coronary heart disease	25 (10.1)	13 (12.9)	4 (12.9)	2 (9.1)	0 (0)	5 (10.9)	1 (5.3)	0 (0)	0 (0)
Venous thromboembolism	7 (2.8)	4 (4.0)	0 (0)	0 (0)	1 (5.6)	0 (0)	2 (10.5)	0 (0)	0 (0)
Other comorbidities	109 (44.0)	46 (45.5)	8 (25.8)	12 (54.5)	7 (38.9)	23 (50.0)	7 (36.8)	3 (50.0)	3 (60.0)

ILD, interstitial lung disease; IQR, interquartile range.

**Table 8 jcm-11-07321-t008:** Characteristics of patients with lung cancer in IIP and non-IIP populations.

Characteristics	Total *n* = 101 (%)	IIP *n* = 41 (%)	Non-IIP *n* = 60 (%)	*p* Value
Age, Median (IQR) years	66 (60–71)	66 (59–72)	66 (60–71)	0.097
Male	76 (75.2)	33 (80.5)	43 (71.7)	0.356
Pathology				0.072
Adenocarcinoma	54 (53.5)	24 (58.5)	30 (50.0)	
Squamous cell carcinoma	29 (28.7)	7 (17.1)	22 (36.7)	
Others	18 (17.8)	10 (24.4)	8 (13.3)	
Laterality				0.344
Left	37 (36.6)	18 (43.9)	19 (31.7)	
Right	59 (58.4)	22 (53.7)	37 (61.7)	
Bilateral	5 (5.0)	1 (2.4)	4 (6.7)	
Comorbidities				
Hypertension	24 (23.8)	8 (19.5)	16 (26.7)	0.480
Diabetes	15 (14.9)	5 (12.2)	10 (16.7)	0.583
Coronary heart disease	13 (12.9)	7 (17.1)	6 (10.0)	0.369
Venous thromboembolism	4 (4.0)	2 (4.9)	2 (3.3)	0.537
Other comorbidities	46 (45.5)	18 (43.9)	28 (46.7)	0.840
ACCI, mean (±SD)	6.5 ± 1.3	6.3 ± 1.2	6.6 ± 1.4	0.278

IIP, idiopathic interstitial pneumonia; IQR, interquartile range; ACCI, age-adjusted Charlson comorbidity index; SD, standard deviation.

## Data Availability

The datasets generated during and/or analyzed during the current study are available from the corresponding author upon reasonable request.

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
