# Peer review of "Malignancies in Patients with Interstitial Lung Diseases: A Single Center Observational Study"

_jcm, 2022, doi:10.3390/jcm11247321_

Round 1

Reviewer 1 Report

Very good idea; the work is well written, in a clear and synthetic way; it has excellent case studies and treats not only lung cancer but also the most important cancers.

However, there are observations and changes that should be made:

1) the quality of the images of the CTs of the chest are not good, as no images with pulmonary filter have been chosen, but so the pathology of the connective is not well exalted

2) there are no images regarding other cancers; at the time other neoplasms are mentioned, it would be appropriate if they were illustrated (at least the most imposed).

3) for such an important job, only 23 bibliography items seem to me very few, they should be implemented.

For the rest, i really like the article.

Reviewer 2 Report

The aims of this study are of interest and is performed on large amount of patients but there are still some questionable issues: 

1. Regarding the statistics there is only a descriptive statistics with no other analysis. Although you mentioned the usage of t-test or Mann-Whitney U test, I did not see the results in the Results section.

2. For comorbidities there are some dedicate scores so you can get further correlations between comorbidities and malignancies. (https://www.mdcalc.com/calc/3917/charlson-comorbidity-index-cci)

3. Although the highest incidence of malignancies are present in OP there is a very few comments at the Discussion.

4. For OP you use organizing pneumonia as well as organized pneumonia. I think you have to choose one of this or to use to different acronyms.

Round 2

Reviewer 1 Report

This version is fine, the deficient points have been adjusted, the captions have been improved and the bibliography has been increased.

Reviewer 2 Report

None